# Multi-objective optimization of nitrile rubber and thermosets modified bituminous mix using desirability approach

Avani Chopra[1], Sandeep Singh[1]*, Abhishek Kanoungo[2], Gurpreet Singh[3], Naveen Kumar Gupta[4], Shubham Sharma[5,6]*, Sanjeev Kumar Joshi[7], Sayed M. Eldin[8]*

1 Department of Civil Engineering, Chandigarh University, Mohali, Punjab, India, 2 Department of Civil Engineering, Chitkara School of Engineering & Technology, Chitkara University, Himachal Pradesh, India, 3 Department of Mechanical Engineering, Chandigarh University, Mohali, Punjab, India, 4 Mechanical Engineering Department, Institute of Engineering and Technology, GLA University, Mathura, UP, India, 5 Mechanical Engineering Department, University Centre for Research and Development, Chandigarh University, Mohali, Punjab, India, 6 School of Mechanical and Automotive Engineering, Qingdao University of Technology, Qingdao, China, 7 Uttaranchal Institute of Technology, Uttaranchal University, Dehradun, India, 8 Center of Research, Faculty of Engineering, Future University in Egypt, New Cairo, Egypt

* elsayed.tageldin@fue.edu.eg (SME); drsandeep1786@gmail.com (SS); shubham543sharma@gmail.com, shubhamsharmacsirclri@gmail.com (SS)

**Data Availability Statement:** All relevant data are within the paper.

**Funding:** The author(s) received no specific funding for this work.

## Abstract

A variety of materials, including waste and rubber products, have been used in road construction to improve the performance of bituminous pavements. The present investigation is focused on modifying bitumen using Nitrile rubber (NBR) with different thermosets namely Bakelite (B), Furan Resin (FR), and Epoxy resin (ER). The emphasis of the problem is to arrive at a mix to achieve maximum Marshall Stability (MS) and minimum flow value of Modified Bituminous Concrete. Taguchi DOE technique has been used to design the experiments using Minitab software. Analysis of Variance (ANOVA) and Multi-objective optimization has been performed using the desirability approach in Design expert software. ANOVA analysis predicts that NBR, B, ER, and FR are the major significant parameters for Marshall Stability (MS) and Flow Value (FV). It has also been analyzed from SEM and EDS images of modified bitumen that sample S1 (5% NBR, 10% Bakelite, 10% FR, 2.5% ER) has a fine surface with small pores as compared to sample S34 (10% NBR, 0% Bakelite 10% FR, 2.5% ER). Multi-optimization results suggested the optimal conditions are achieved at NBR-7.6%, Bakelite-4.8%, FR-2.5%, and ER-2.6% for MS and FV. The maximum MS is 14.84 KN and the minimum FV is 2.84 mm is obtained using optimum conditions. To validate the optimization results, the confirmation runs have been conducted, and obtained results are within 5% error with optimal conditions.

## 1. Introduction

There has been a phenomenal increase in road traffic and this has led to an increase in the road network in both developing and developed countries. Thus, a need has arisen to explore

**Competing interests:** The authors have declared that no competing interests exist.

the potential for utilizing waste products in order to reduce the consumption of bitumen for sustainable pavements. A variety of additives have been developed to improve the performance of bitumen which have been used successfully in many applications [1]. To boost bitumen performance, modifiers and additives such as polymers, oxidants, antioxidants, chemical modifiers, expanders, anti-stripping additives, and hydrocarbons have been used in pavements [2]. The addition of polymers along with bitumen forms a chain of small molecules which results in enhanced performance of the pavement. The polymer-modified bitumen increases the strength against rutting, fatigue, and cracks. It also increases elastic recovery, viscosity, softening point, adhesion, and flexibility [3, 4]. Moreover, modified bitumen improves the durability and breakdown due to rutting deformations in high temperatures and thereby increases the fatigue life. Modified bituminous mix has become appropriate for being used as a wearing or binder course due to enrichment in the stiffness modulus, creep resistance, and indirect tensile strength which the road surface due to heavy traffic. Therefore, modified bitumen can be effectively utilized at busy intersections, bridge decks, and roundabouts to increase the life of the road surfacing.

Over the past fifteen years, the use of modified bitumen has increased to almost 05 million tons, with crumb rubber-modified bitumen accounting for nearly 80%. The remaining 20% is consumed through the use of polymer-modified bitumen. The use of Thermosetting asphaltic mixtures including bitumen modified with polymers results in increasing the adhesive as well as resistance properties, fatigue performance, and rigidity modulus [5].

The researchers have made many efforts to explore the potential of using different types of rubbers and thermosets to enhance the performance of pavements as an alternative road construction material. Bitumen modification using crumb rubber (CR) with up to 7% content leads to enhancement in the mechanical properties of the bituminous mix. The penetration test revealed that by increasing the amount of crumb rubber a reduction of 7.54 mm of the depth of penetration was observed which evidently reveals better resistance to asphalt binder deformation thereby making it more useful for hot climates [6]. Joohari and Giustozzi [7] investigated the effect of hybrid PMB blends by utilizing CR along with styrene–butadiene–styrene (SBS) and ethylene-vinyl acetate (EVA) on physical properties of bitumen which showed a decrease in penetration value, while the softening point increased for all PMB. The addition of 2% EVA to the CR/SBS blend has also proven to provide stiffness at intermediate and high temperatures. Chinoun et al. [8] employed nitrile butadiene rubber (NBR) waste / Ethylene vinyl acetate (EVA) association with 5% NBR/EVA, which led to decreased penetration, indicating an increase in modified bitumen hardness. Zhang et al. [9] used Polyurethane (PU) to modify polyester resin (UPR) which is a thermosetting material that resulted in improved strength by 12% and a marginal improvement in the low-temperature performance. Saha & Suman S. [10] investigated the effect of 60/70 penetration grade bitumen modified with the addition of 1% to 5% Bakelite (B) on mechanical and qualitative properties. In the softening stage, viscosity, and binder penetration enhanced and the Marshall Stability value improved with a 2% bakelite addition. Cubuk et al. [11] examined the use of phenol-formaldehyde on bitumen modification and reported that this modified mix increased durability against high traffic and environmental dangers due to raised flexibility as well as stripping qualities at 2% content of phenol-formaldehyde. Bostancioglu and Oruc [12] tested the Marshall properties of modified asphalt mixes made by combining activated carbon (CA) and furan resin (FR). The results reported that the addition of CA and FR substantially enhanced the Marshall's stability value by 9%, and due to the addition of FR the Marshall quotient increased by 25%. In the case of the stiffness modulus of indirect tensile, the mixtures containing 5% FR and 10% CA had an overall stiffness modulus of 16% higher than the conventional combination. The usage of Furan Resin content between 2% - 6% in 50/70 penetration grade

bitumen enhances the aging tolerance and consistency of bitumen and minimizes the humidity and temperature sensitivity [13]. Recently chemical additives are being used for increasing the efficacy of recycled asphalt mixtures. Bituminous mix treated with epoxy resin (ER) mixture, SBR Styrene-Butadiene rubber (SBR), and PVA resulted in enhanced pavement performance towards acid resistance resulting in fuel efficiency. The dynamic residue viscosity test revealed a potentially large viscosity value for several applications of pavement [14]. Zhang et al. tested the efficiency of modified bitumen (tack coat) emulsion which was modified with aqueous ER and SBR. The inclusion of SBR and ER increased the shear strength, as well as the high and low-temperature characteristics [15]. The use of asphalt and bitumen using modified polymers, such as ER, revealed that ER-containing asphalt pavements have improved riding capabilities. A considerable jump in the asphalt pavement strength is observed when 20–35% of bitumen is replaced with ER, especially at high temperatures. Chopra and Singh [16] inferred that by modifying bitumen with 3.25% ER the mix the softening point was reported as the 59°C which is higher than that of the unmodified bitumen. The test carried out also clearly indicates that replacing bitumen with epoxy reduces the penetration and ductility values, but increases the softening point and specific gravity of the mix. Ji et al. [17] investigated the quality of cold-mix asphalt made with modified bitumen using waterborne epoxy. The micro-surfaces created with aqueous epoxy resin-modified bitumen emulsion also demonstrated promise resistance to moisture damage, skid, and rutting.

The optimum percentage of modifiers is essential for improving pavement performance characteristics. Various researchers utilized different tools and techniques for the optimization of the Bituminous Mixes for better pavement performance and pavement design. Gundaliya & Patel [18] utilized Genetic Algorithm (GA) to determine the optimum mix for Bituminous Concrete. Tsai et al. [19] obtained satisfactory results by utilizing GA to solve non–linear optimization problems related to pavement design. Yu et al. [20] developed a model for multi-objective optimization to solve the three decision objectives maintenance arrangement of asphalt pavement. Nik et al. [21] used different combinations of particle swarm optimization (PSO) and hybrid genetic algorithm to suggest an optimal solution for surveyed pavement inspection units (SIUs) to reduce cost, improve the accuracy of pavement network, and minimization of inspection errors. Santos et al. [22] proposed a new Adaptive Hybrid Genetic Algorithm (AHGA) to enhance the effectiveness and overall efficiency of the search. results revealed that the proposed AHGA statistically performed better than the traditional GA in efficiency terms. Sebaaly et al. [23] utilized GA to develop models for Marshall mix design for the four asphalt mix variables, i.e., flow, stability, theoretical maximum specific gravity ($G_{mm}$), and bulk specific gravity ($G_{mb}$). Liu et al. [24] did multi-response optimization using Response Surface Method (RSM) to achieve satisfactory properties in the modified asphalt binder. Bala et al. [25] used Response Surface Methodology (RSM) for the prediction of Marshall volumetric properties of modified bitumen containing polyethylene and nano-silica. Moghaddam et al. [26] used the point prediction function in the Design-Expert software for analyzing the optimal values of polyethylene terephthalate (PET) and asphalt contents to meet the design requirements of Stone Mastic asphalt (SMA) mixtures. Desirability functions are suitable for obtaining optimal blend compositions thereby lowering the usage of virgin bitumen for bituminous modifications. RSM helps in the prediction of responses on the cost economy for the Polymer Modified Bitumen (PMBs) for the varied modifier compositions [27]. Hamza et al. [28] developed a model using RSM for predicting the value of Optimum Binder Content (OBC) for various levels of compaction temperature and Recycled Asphalt (RA) content. RSM was used for determining the optimal amount of polyurethane modifier in the bitumen by central composite design [29].

The Bakelite, NBR, FR, ER, and other modifiers have been used to prepare the modified bituminous mix and their characteristics were analyzed for performance improvement. Most

researchers have used a single or combination of two modifiers in a modified bitumen mix in the reported literature. But the combination using four different modifiers (NBR, B, FR, ER) has been rarely used in modified bitumen mix. To achieve the economy and improve the characteristics, optimization of the modified bitumen mix is very essential. In the present work, Multi-objective optimization of modified bitumen mix has been attempted for the Marshall stability (MS) and flow value (FV) using the desirability function approach in design expert software.

## 2. Materials and methods

To prepare the modified bitumen, various materials such as bitumen, aggregates, fillers, NBR, Bakelite, FR, and ER have been used in the present work. These materials along with their properties are discussed in detail.

### 2.1 Materials and their properties

**2.1.1 Bitumen.** Bitumen grade VG-40 was used in this research work to study the Volumetric and Marshall properties of bitumen modified with the addition of Nitrile Rubber and Thermosets (Bakelite, FR, and ER). The physical properties of bitumen are given in Table 1.

**2.1.2 Aggregates.** The crushed coarse and fine aggregates were used and several physical tests like Los Angles Abrasion test, water absorption test, aggregate impact, and crushing value tests were performed. Fig 1 shows the gradation limits of the aggregates used in this study matching Grade 1 of Bituminous Concrete as per MORT&H.

**2.1.3 Filler.** Stone dust has been used as the filler for preparing the conventional and modified bituminous mix. The filler should be classified within the limits set out in Table 500–9 of MORT&H specifications mentioned in Table 2.

**2.1.4. Nitrile rubber.** Nitrile Rubber is the acrylonitrile butadiene rubber that is, obtained from the milling shoe soles and industrial waste blackish in color [8, 30, 31]. Table 3 showcases the properties of NBR as obtained from the vendor. Five characterization tests were performed on the nitrile rubber samples. The values given above are compared with the American Society of Testing and Material (ASTM) i.e., ASTM D2000, ASTM D297, ASTM D573, and the test values were found to be well within the limits.

**2.1.5. Bakelite.** Bakelite is the trade name given to phenol-formaldehyde which is a thermosetting plastic. Table 4 shows the properties of Bakelite. As per IS 2036–1995 Grade P1 the tensile strength and other tests were performed on the Bakelite sample. It has been observed that the obtained value of tensile strength, flexural strength, and specific gravity were well within the laid down values as per IS 2036–1995 Grade P1 [32–34].

**2.1.6. Furan resin.** Furan is a furfural-derived thermoset resin. Table 5 shows the properties of the Furan resin sample used in this investigation. The properties were checked through experiments and then compared with the laid down corresponding limits as mentioned in IS 4832 Part II. The flexural strength, compressive strength, and water absorption by weight in percentage were found to be well within the laid own limits and hence it was used in the present study [35–37].

**Table 1. Physical properties of bitumen grade VG-40.**

| Property of bitumen | Test value | Range as per IS 73–2013 |
|---|---|---|
| Penetration value (mm) | 49 | Minimum 35 |
| Softening point (˚C) | 53 | Minimum 50 |
| Ductility value (cm) | 29 | Minimum 25 |
| Flash Point (˚C) | 228 | Minimum 220 |

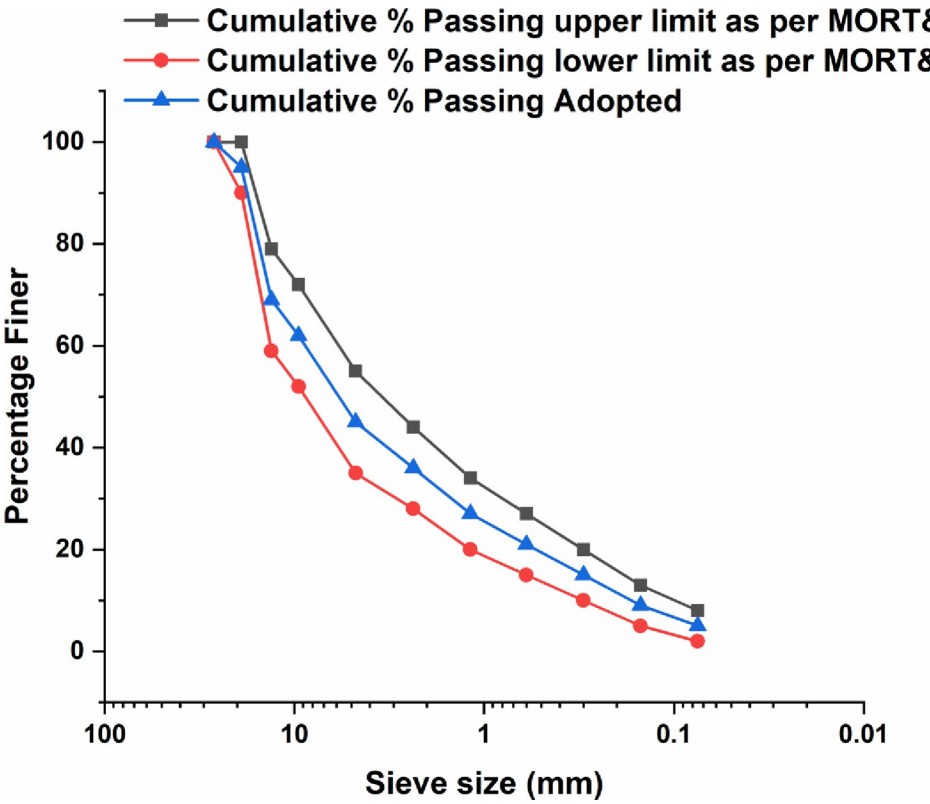

**Fig 1. Gradation of aggregates used in this study.**

**2.1.7. Epoxy resin.** Epoxy resin is a flexible usually thermosetting resin made by copolymerization of an epoxide with another compound having two hydroxyl groups and used chiefly in coatings and adhesives [29]. Table 6 shows the properties of the Epoxy resin sample obtained from the supplier and it was compared with the values as laid down in IS 9197. It has been observed that the critical properties were well within the limits laid down in IS 9197 specifications.

To determine the properties of materials numerous tests were conducted and a bituminous mix was prepared, which was then subjected to Marshall Tests for ascertaining the optimum content of bitumen within the prescribed limits as per MORTH. The research methods and flow to conduct the laboratory work are mentioned in Fig 2.

## 2.2 Sample preparation

The above-discussed materials were used to prepare the modified bituminous mixes for the evaluation of Marshall Properties. Aggregates and filler weighing 1200 g are heated at 160–

**Table 2. Gradation of filler.**

| IS Sieve (mm) | Cumulative Percent Passing by weight of Total Aggregate | Range as per MORT&H for Cumulative Percent Passing by weight of Total Aggregate |
|---|---|---|
| 0.6 | 100 | 100 |
| 0.3 | 95 | 95–100 |
| 0.075 | 90 | 85–100 |

**Table 3. Properties of nitrile rubber.**

| Property Name | Value |
|---|---|
| Hardness | 70+/-5 shore |
| Modulus of elasticity | Greater than or Equal to11MPa |
| Tensile strength | Greater than or Equal to17MPa |
| Density | 1.28 g/cc |
| Temperature Range | -30 to110˚C |

175˚C. Bitumen is heated at 160–170˚C and added in predetermined percentages. The mixing temperature is maintained at 160–170˚C. In the pre-heated mould, the mix is poured and compaction is done up to 75 blows on both sides of the specimen by the rammer. The weight of mixed materials used in the specimen preparation can be adjusted to obtain 63.5+/-3 mm thickness. In the next trial, increase the bitumen concentration by 0.5%, and retesting was done.

## 2.3 Determination of Optimum Bitumen Content (OBC)

In order to determine the OBC content, the raw bitumen VG-40 was taken for experimentation. The properties of the mix were determined based on the Marshall Mix design. The OBC was analyzed based on the behavior of the mix by comparing it with stability, flow, air voids, Percent voids filled with bitumen (VFB) &Percent void in mixed aggregate (VMA) &. In order to achieve this, the specimens were prepared to have binder content varying from 5.0 to 6.5% by weight with an increment.

The maximum theoretical specific gravity obtained by an asphalt density tester based on ASTM D 2041 determines the maximum specific gravity of the mixture. To get the stability and flow value of specimens, the Marshall Stability test was conducted based on ASTM D 6927 for BC mixtures. The recommended value for determination of OBC is corresponding to 4% Air Voids, after checking the values of the above-discussed properties to fall within the limits of MORT&H Specifications was achieved at 5.56% of bitumen. Table 7 shows the values of the other volumetric properties which are being accepted as per the MORT&H limits.

## 2.4 Marshall mix design

Marshall stability test estimates how well the Marshall mix design sample will perform at the loading rate of 50.8 mm/min, the measurement of the largest load supported by the test specimen is measured which is defined as stability. Another dial gauge measures the deformation caused by the loading which indicates the flow value.

Modifiers in various combinations can be added in bitumen either by using the Dry or Wet Process. In the present work, the wet process was adopted to ensure homogeneous mixing and blending [38, 39]. The modifiers along with bitumen were mixed by hand for 15 min and

**Table 4. Properties of bakelite.**

| Property Name | Value | Values as per IS 2036 |
|---|---|---|
| Tensile strength | 1050 kg/cm$^2$ | 1050 kg/cm$^2$ |
| Density | 1.3 g/cc | 1.3 g/cc |
| Flexural Strength | 1350 kg/cm$^2$ | 1300 kg/cm$^2$ (Min) |
| Shear Strength | 800 kg/cm$^2$ (750 Min) | 750 kg/cm$^2$ (Min) |
| Specific Gravity | 1.38 | 1.38 |

**Table 5. Properties of furan resin.**

| Property Name | Value | Values as per IS 4832 Part II |
|---|---|---|
| Colour | Colourless | - |
| Flexural Strength | 80 kg/cm$^2$ | Min 75 kg/cm$^2$ |
| Compressive strength | 370 kg/cm$^2$ | Min 350 kg/cm$^2$ |
| Absorption by weight in percentage | 0.68% | Max 1% |
| Boiling point | 313˚C | - |

heated at 160 to 170˚C before mixing with preheated aggregates (160 to 175˚C). Thereafter the aggregates along with modified bitumen were transferred in a preheated mould for Marshall Testing.

## 2.5 Design of Experiment (DOE)

In this experimentation, the Taguchi DOE technique is used to design the experiments. It has the advantage over the traditional DOE method as it can consider multiple factors at a time to determine the optimal parameters by using fewer experimental runs than the traditional DOE method [37–39]. To perform the investigation, the approach using the L81 orthogonal array was selected. In this method, a total of 243 experiments have been conducted for studying the complete space using the L81 orthogonal array.

The independent variables are NBR, B, FR, and ER content, while the responses considered, are MS and FV. The preliminary experimentation and reported literature were used to select the independent variables and their range. Table 8 shows the different input parameters with the selected levels for the experimentation. The experiments have been conducted as per the Design of Experiment approach using Mini-tab to arrive at the Marshall Properties of the modified bituminous mix as shown in S1 Table in S1 Appendix.

## 4. Results and discussion

The obtained experimental results have been analyzed using Analysis of variance (ANOVA) [38, 39]. Significant quadratic models and regression equations have been developed for MS and FV. S2 and S3 Tables in S1 Appendix represent the ANOVA results for Marshall Stability and Flow respectively. The effect of varying the percentage of NBR, Bakelite, FR & ER with 95% confidence intervals using one factor at a time approach are reported in Figs 3 and 4. Fig 4 clearly represents that MS is increasing with the increase in the percentage of NBR, Bakelite, and FR. The drop in MS is also analyzed by increasing the percentage of FR beyond 5% value. Fig 4 demonstrates that FV is slowly increasing with the percentage of NBR but a rapid increase in FV is observed by increasing the percentage of FR to 5%. Bakelite has a small impact on the FV, initially starts decreasing slowly till 5%, and afterward, FV starts increasing. On the other side, FV is rising by increasing the percentage of ER to 5% and afterward starts declining as shown in Fig 4.

**Table 6. Properties of epoxy resin.**

| Property Name | Value | Values as per IS 9197 |
|---|---|---|
| Colour | Colourless | - |
| Viscosity at 27˚C | 11 Pa s | 3 to 20 Pa s |
| Specific gravity at 27˚C | 1.09 | 0.5 to 1.20 |
| Hydrolyzable Chorine percentage | 0.40 | Max 0.60 |
| Boiling point | 178˚C | - |

| Collection of Materials and Modifiers i.e., bitumen, aggregates, NBR and Thermosets |
|---|

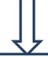

| Testing of Bitumen i.e., Softening Point, Penetration Value and Aggregates i.e., Flakiness Index, Water Absorption, Los Angeles Value |
|---|

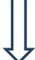

| Preparation of Marshall Sample and Determination of Optimum Bitumen Content |
|---|

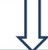

| Design on modified bituminous mixes and determination of Marshall Stability and Flow values. |
|---|

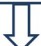

| Blend Characterization by SEM (Scanning Electron Microscopy), EDS (Energy-dispersive X-ray Spectroscopy) |
|---|

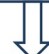

| ANOVA Analysis and Multi objective optimization using desirability approach in design expert software |
|---|

**Fig 2. Research methodology to conduct laboratory work.**

Fig 5 represents that there is good agreement between projected and actual response levels. All models can adequately be employed effectively to determine the optimal condition. To analyze the interaction effects of NBR, B, FR, and ER on MS and FV, 3D interaction plots were generated using Design Expert software. Fig 6 shows the interaction 3D plots for MS and FV.

Fig 6(A) represents that the maximum stability is attained at 10% NBR and 10% Bakelite. The stability continuously increases by increasing the percentage of NBR. The main reason is that NBR is a softer material thus it has more flexibility. Also, as the B content increases

**Table 7. Volumetric properties of bituminous mix at OBC of 5.56%.**

| S. No. | Properties | Value Obtained | Permitted Values as per MORT&H Table-500-17 |
|---|---|---|---|
| 1 | Stability (kN) | 9.91 | 9 (minimum) |
| 2 | Flow (mm) | 2.96 | 2–4 |
| 3 | Marshall Quotient (MQ) | 4.27 | 2–5 |
| 4 | % Air Voids (VA) | 4 | 3–5 |
| 5 | % Voids Filled with Bitumen (VFB) | 70.01 | 65–75 |
| 6 | % Voids in Mineral Aggregate (VMA) | 12.98 | 12 (minimum) |
| 7 | % Tensile Strength Ratio (TSR) | 80.55 | 80 (minimum) |

**Table 8. Level of Variables used during experimentation.**

| S.No. | BLEND PARAMETERS | LEVEL-1 | LEVEL-2 | LEVEL-3 |
|---|---|---|---|---|
| | | 1 | 2 | 3 |
| 1 | A-NBR (%) | 0 | 5 | 10 |
| 2 | B-Bakelite (%) | 0 | 5 | 10 |
| 3 | C-FR (%) | 0 | 5 | 10 |
| 4 | D-ER (%) | 0 | 2.5 | 5 |

beyond 5% there is a marginal dip in the Stability value as compared with the peak Stability. After 5% B level due to the combined effect of hardness and fragility the stability increases. This showed that the hardness of Bakelite and flexibility of NBR complemented to each other resulting in higher stability. Fig 6(B) shows that the maximum stability is attained at 10% NBR and 5% FR levels. The stability continuously increases by increasing the percentage of NBR. The main reason is that NBR is a softer material thus it has more flexibility. On the other side, the stability increased initially by increasing the Furan Resin content but after the FR content was increased beyond 5% the stability of the mix decreased. The main reason is that FR has a

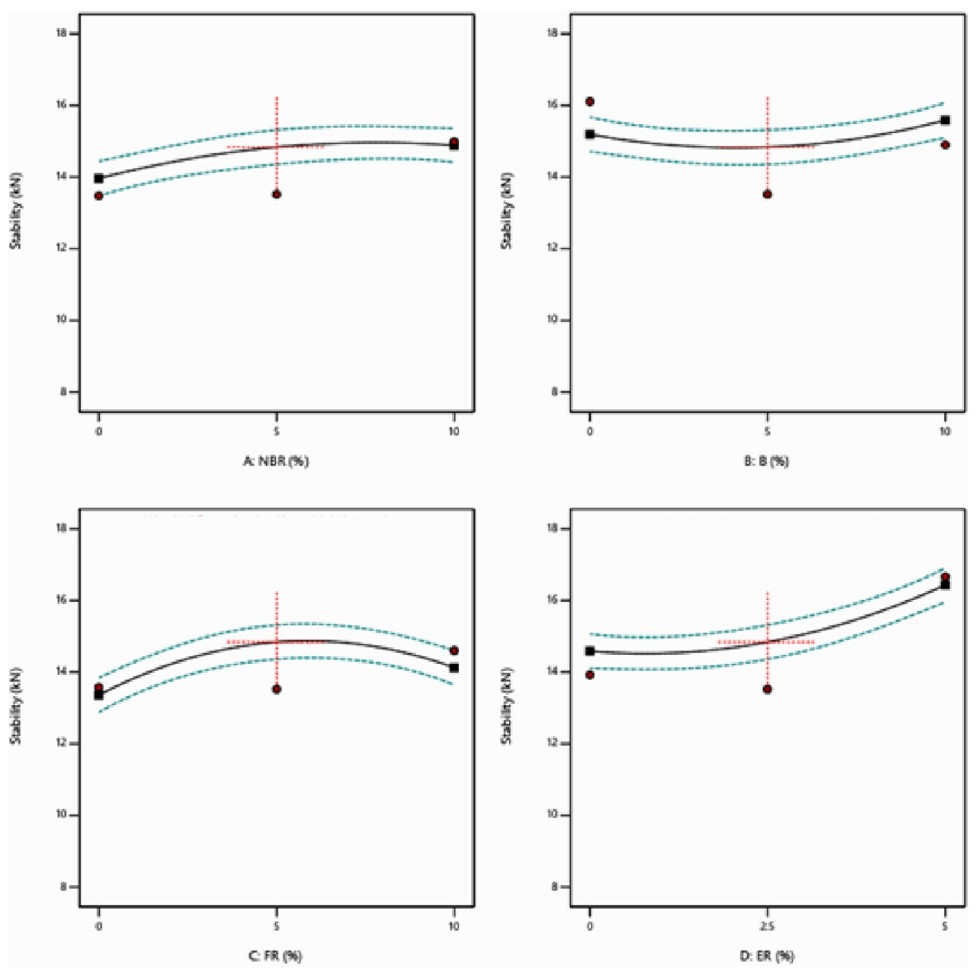

**Fig 3. Individual effect plot for stability.**

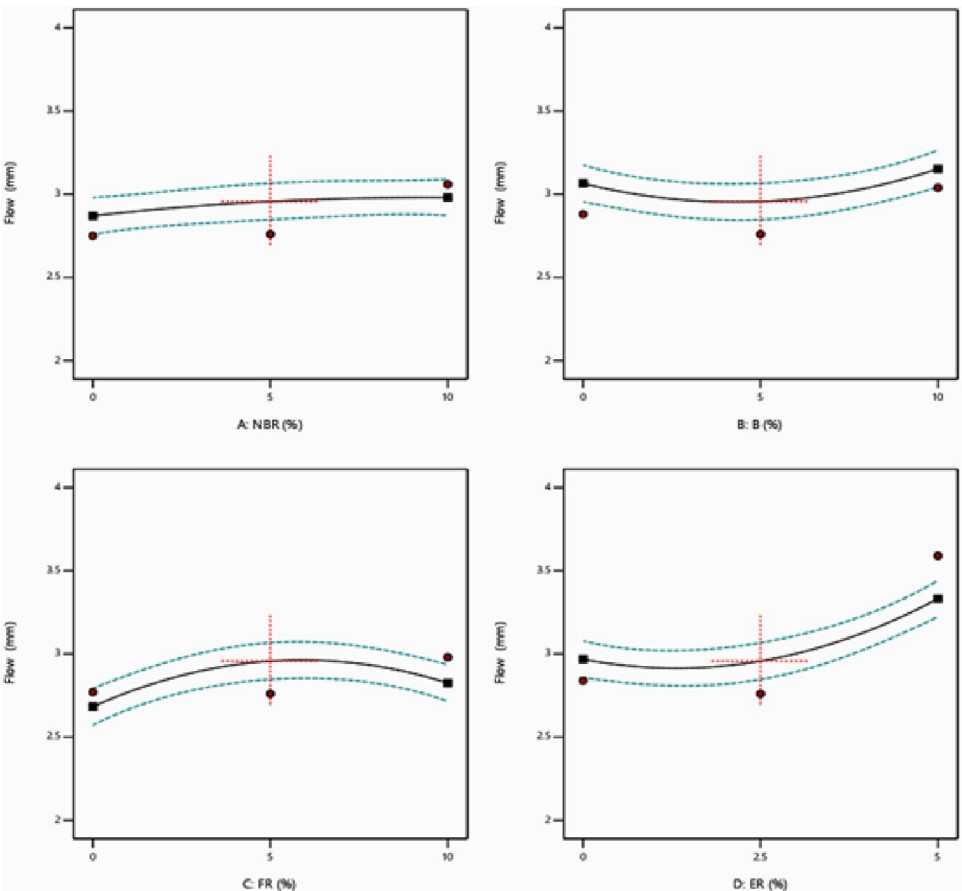

**Fig 4. Individual effect plot for flow value.**

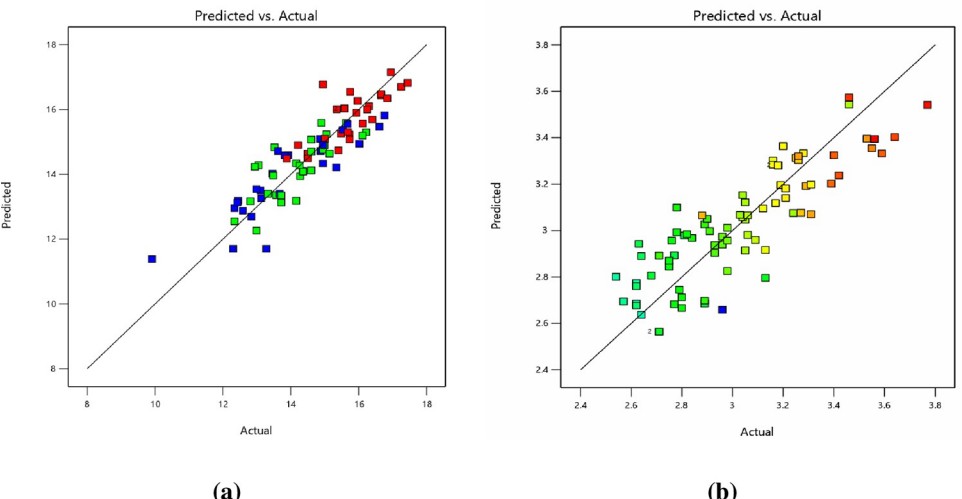

**(a)**　　　　　　　　　　　　　　　　**(b)**

**Fig 5.** Predicted vs Actual value plot (a) Marshall Stability; (b) Flow Value.

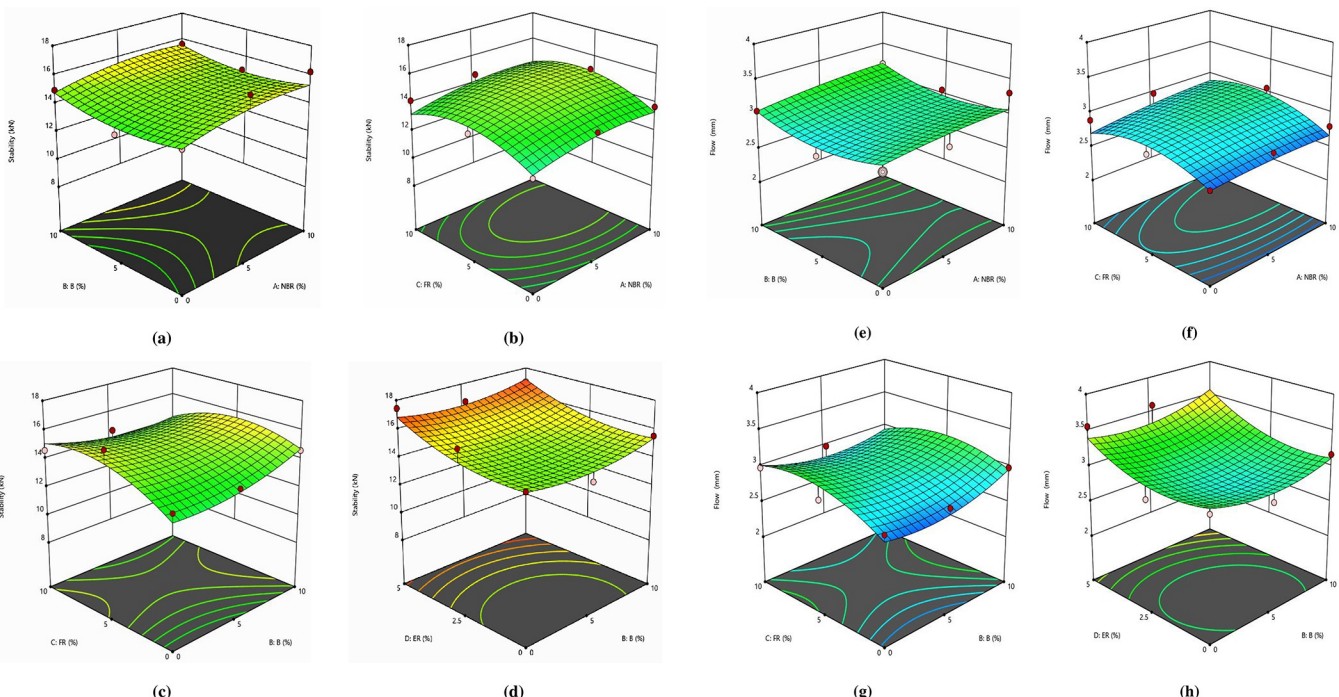

**Fig 6.** (a) Effect of B and NBR on Stability; (b) Effect of FR and NBR on Stability; (c) Effect of FR and B on Stability; (d) Effect of ER and B on Stability (e) Effect of B and NBR on Flow; (f) Effect of FR and NBR on Flow; (g) Effect of FR and B on Flow; (h) Effect of ER and B on Flow.

comparatively stronger bond amongst its molecules but when it is mixed with NBR the bond tends to become less adhesive and hence the stability decreases.

Fig 6(C) shows that the maximum stability is attained at 10% B and 0% FR levels. The stability continuously increases by increasing the percentage of FR up to 5% and thereafter the stability started decreasing. The main reason is that FR is having a stronger bond but the addition of Bakelite which is a harder and brittle material makes the mix stability value lesser. This is attributed to the fact that Bakelite is a comparatively strong material and hard and brittle hence its addition to the mix makes it less adhesive leading to a decrease in stability.

Fig 6(D) shows that the maximum stability is attained at 5% ER and 10% B level. The stability continuously increases by increasing the percentage of ER and the Bakelite percentage. The main reason is that ER is having a stronger bond but with the addition of Bakelite which is a harder and brittle material, has less flexibility hence the stability decreased. Fig 6(E) represents that the minimum flow is attained at 5% NBR and 5% Bakelite. The flow continuously increases by increasing the percentage of NBR. The main reason is that NBR is a softer material making the mix more flowable. Increased levels of Bakelite make the mix stiff thereby reducing the flow value.

Fig 6(F) shows that minimum flow is obtained at 5% NBR and 0% FR levels. As the NBR content is increased in the mix the flow value increase which is attributed to the flowable property associated with rubber. At equal percentages of NBR and FR that is 5% the flow is maximized. As the level of FR is further increased to 10% there is a marginal dip in the flow value due to proper binder coating on the surface of the aggregate particles enforcing stiffness in the mixture. Fig 6(G) represents that minimum flow is obtained at 5% B and 10% FR levels. At 5% B level. On varying the FR level from 0 to 10% the flow value increases to equal levels of B and FR i.e., 5% each, and thereafter the flow value again decreases marginally. The flow indicates the resistance to distortion and a very high value of flow is not desirable. It is found that as the

Bakelite was increased the flow started decreasing rapidly which is attributed to the fact that the stiffness of the mix is enhanced due to the addition of Bakelite. Fig 6(H) shows that minimum flow is obtained at 5% B and 2.5% ER levels. The flow continuously increases by increasing the percentage of ER from 0 to 5%. At 2.5% ER level and on increasing the B level to 5% the flow value is reduced. The main reason is that ER is having a stronger bond but with the addition of Bakelite which is a harder and more brittle material thus, it has less flexibility hence the flow decreased.

## 5. Surface morphology

To study the microstructure and chemical composition of modified bitumen, Scanning Electron Microscopy (SEM) has been utilized. The SEM and EDS analysis were performed on equipment Model: JEOL JSM-6610LV which has the Resolution in High vacuum Mode: 3nm, Magnification: Up to 300000, Vacuum Pressure in Chamber: Adjustable 10–270 Pa, and Accelerating Voltage: 0.3–30 kV.

For SEM and EDS analysis, the two best samples S1 and S34 were selected on the basis of the highest Marshall Quotient (MQ) values. MQ is the ratio of MS and FV. High MQ values indicate a mix with high stiffness and with a greater ability to spread the applied load and resistance to creep deformation [40–44]. We have also achieved a marked increase in the MQ value for S1 is 4.96 and for S34 is 4.95, which reflects an improvement of the mechanical performances of waste NBR and thermosets modified bituminous mix.

SEM and EDS images of modified bitumen samples S1 and S34 at magnification X50 at 100μm scale and magnification X100 at 100μm are shown in Fig 6. It is clearly seen from Fig 7 (A) that a fine surface with small pores is visible in the SEM image of modified bitumen sample S1 due to the presence of Bakelite, NBR, ER, and FR. Fig 7(B) represented the EDS of modified bitumen sample S1 and respective peaks value in the table showing the maximum value of carbon value at 91.93% and Sulphur value at 4.37% rather than Nitrogen at 3.70% respectively.

It has been observed from Fig 7(C) that the fine flaky shapes with small pores or large pores are visible on the surface of modified bitumen sample S34 due to the absence of Bakelite and the presence of NBR, FR, and epoxy resin. Fig 7(D) represented the EDS of modified bitumen sample S34 and respective peaks value in the table showing the maximum value of carbon at 88.23%, Oxygen value at 6.30%, Sulphur value at 2.54%, Calcium value at 1.70%, Nitrogen at 1.23% respectively.

## 6. Multi-objective optimization

Optimization is a very essential criterion to enhance the modified mix properties [45–48]. The optimum parametric settings of MS adversely impact the FV and similarly the optimum parametric settings of FV adversely impact MS. Thus, there is a necessity to determine a unique set of parameters that provides acceptable results for the values of MS and FV. The desirability approach using design expert software has been utilized to optimize the input parameter values for MS and FV.

### 6.1 Regression analysis

To carry out the regression method analysis Design Expert software was used to investigate the impact of input factors on the output factors. Regression equations were generated for each output response MS and FV which are represented by Eqs (1), and (2). The main objective is to identify a single set of Marshall variables within set limits, which would enhance the MS and minimize the FV. At first, two equations for both objective functions (MS and FV) are changed

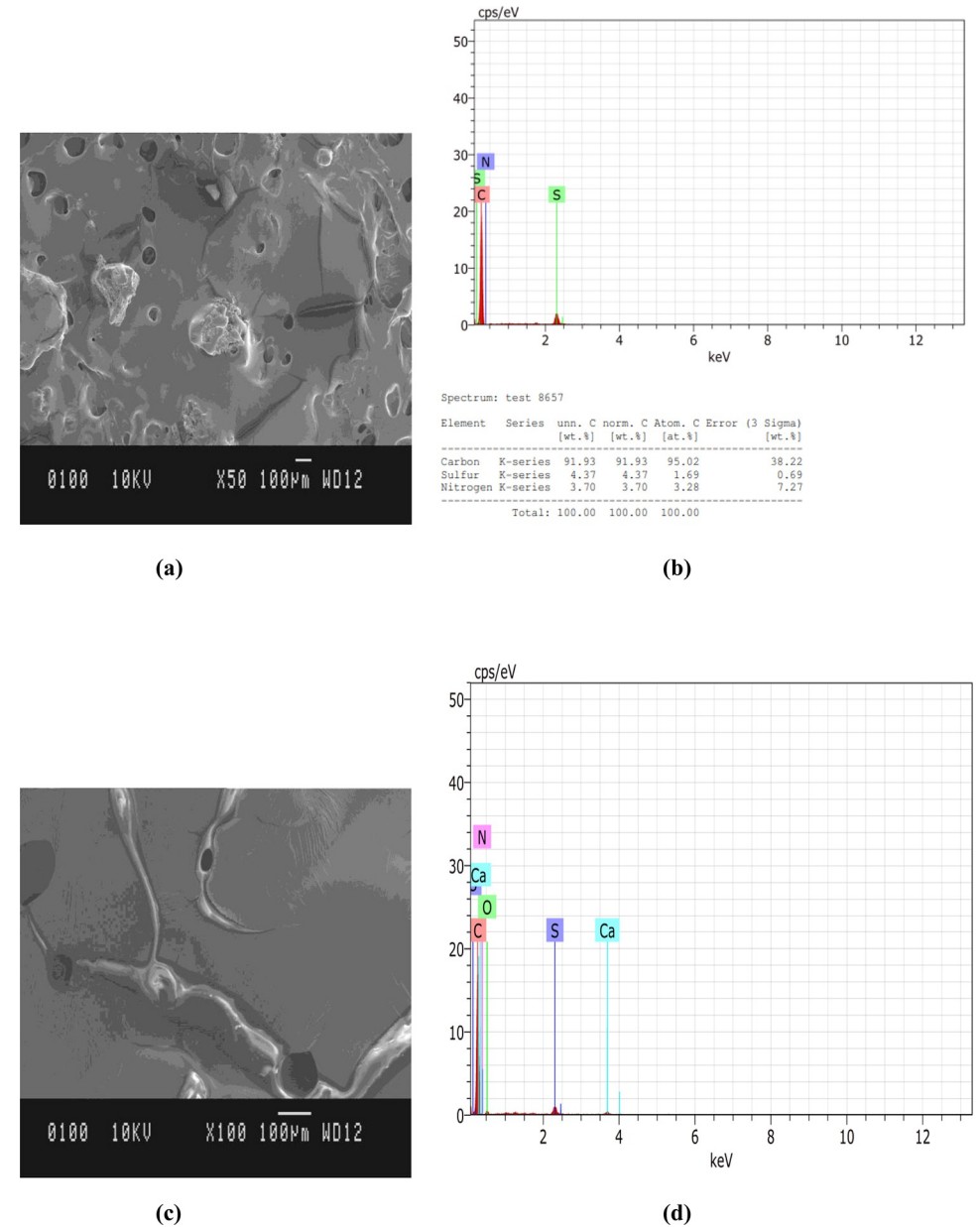

**Fig 7.** Modified bitumen sample S1 (a) SEM image at magnification X50; (b) EDS; (c) SEM image of sample S34 at magnification X100; (d) EDS of sample S34.

over into a single equation, as represented in (3).

$$
\begin{aligned}
MS_{MAX} = {} & +11.38349 + .344750NBR - 0.047712B + 0.650941FR + 0.083228ER \\
& - 0.002078(NBR)(B) + 0.002469(NBR)(FR) - 0.035880(NBR)(ER) \\
& - 0.023691(B)(FR) - 0.002573(B)(ER) - 0.011912(FR)(ER) - 0.016445(NBR) \\
& \times (NBR) + 0.022218(B)(B) - 0.043877(FR)(FR) + 0.107396(ER)(ER
\end{aligned}
\tag{1}
$$

$$FV_{MIN} = +2.65864 + 0.026665NBR - 0.049292B + 0.104669FR - 0.066393ER \\ + 0.000767(NBR)(B) + 0.001627(NBR)(FR) - 0.006030(NBR)(ER) \\ - 0.003139(B)(FR) + 0.003635(B)(ER) - 0.000648(FR)(ER) - 0.001248(NBR) \\ \times (NBR) + 0.006083(B)(B) - 0.008122(FR)(FR) + 0.030880(ER)(ER) \quad (2)$$

Multi-objective Equation using equal weights:

$$Z_{MIN} = -0.5 * (+11.38349 + .344750NBR - 0.047712B + 0.650941FR + 0.083228ER \\ - 0.002078(NBR)(B) + 0.002469(NBR)(FR) - 0.035880(NBR)(ER) - 0.023691(B) \\ \times (FR) - 0.002573(B)(ER) - 0.011912(FR)(ER) - 0.016445(NBR)(NBR) \\ + 0.022218(B)(B) - 0.043877(FR)(FR) + 0.107396(ER)(ER)) + 0.5 * (+2.65864 \\ + 0.026665NBR - 0.049292B + 0.104669FR - 0.066393ER + 0.000767(NBR)(B) \\ + 0.001627(NBR)(FR) - 0.006030(NBR)(ER) - 0.003139(B)(FR) + 0.003635(B) \\ \times (ER) - 0.000648(FR)(ER) - 0.001248(NBR)(NBR) + 0.006083(B)(B) \\ - 0.008122(FR)(FR) + 0.030880(ER)(ER)) \quad (3)$$

## 6.2 Desirability function approach

The desirability function approach involves performing the multi-response optimization in two steps.

i. Determining whether the responses are desirable.

ii. The enhancement of desirability and the discovery of ideal values.

The Design Expert software developed a number of alternatives for this desirability approach, and the solution with the highest level of desirability was chosen. The MS is maximised and the FV is reduced using the desirability strategy. Table 9 represents the constraints utilized for process parameter optimization.

Table 10 represents multi response optimization solutions to obtain maximum MS and FV. The solution with maximum desirability is selected as a best solution. As per selected solution 1, optimal input conditions are A-NBR content 7.57%, B- Bakelite content 4.79%, C-FR content 2.46% and D- ER content 2.58%.

Fig 8 represents the composite desirability of MS and FV. Design expert software was used to create the composite desirability plots. It has been analyzed from graph that the value of composite desirability is 0.66 which predicts the value of output responses for multi-response optimization. The predicted value for MS is 14.48 kN and FV is 2.88 mm after the multi-response optimization.

**6.3 Conformation tests.** In order to justify the optimization results the conformation test has been conducted. The conformation test has been carried out at optimum solutions to validate the characteristics for $MS_{MAX}$ and $FV_{MIN}$. The results of multi-objective optimization advised to set the value of NBR 7.6%, B 4.8%, FR 2.5% and ER 2.6% to get optimum values of

**Table 9. Constraints utilized for multi response optimization.**

| Name | Goal | Lower Limit | Upper Limit | Lower Weight | Upper Weight | Importance |
|------|------|-------------|-------------|--------------|--------------|------------|
| A:NBR | is in range | 0 | 10 | 1 | 1 | 3 |
| B:B | is in range | 0 | 10 | 1 | 1 | 3 |
| C:FR | is in range | 0 | 10 | 1 | 1 | 3 |
| D:ER | is in range | 0 | 5 | 1 | 1 | 3 |
| MS | maximize | 9.91 | 17.44 | 1 | 1 | 3 |
| FV | minimize | 2.54 | 3.77 | 1 | 1 | 3 |

**Table 10. Multi response optimization solutions.**

| Number | NBR | B | FR | ER | MS | FV | Desirability | Remarks |
|---|---|---|---|---|---|---|---|---|
| 1 | 7.573 | 4.793 | 2.463 | 2.581 | 14.482 | 2.879 | 0.663 | Selected |
| 2 | 7.568 | 4.798 | 2.439 | 2.593 | 14.479 | 2.879 | 0.663 | |
| 3 | 7.567 | 4.761 | 2.468 | 2.584 | 14.482 | 2.879 | 0.663 | |
| 4 | 7.521 | 4.810 | 2.453 | 2.579 | 14.480 | 2.879 | 0.663 | |
| 5 | 7.545 | 4.815 | 2.501 | 2.560 | 14.489 | 2.881 | 0.663 | |
| 6 | 5.974 | 2.557 | 10.000 | 2.435 | 14.523 | 2.891 | 0.662 | |
| 7 | 6.016 | 2.557 | 10.000 | 2.426 | 14.524 | 2.891 | 0.662 | |
| 8 | 6.004 | 2.549 | 9.998 | 2.442 | 14.529 | 2.892 | 0.662 | |
| 9 | 5.988 | 2.597 | 9.991 | 2.437 | 14.520 | 2.890 | 0.662 | |
| 10 | 5.982 | 2.586 | 9.979 | 2.415 | 14.519 | 2.890 | 0.662 | |

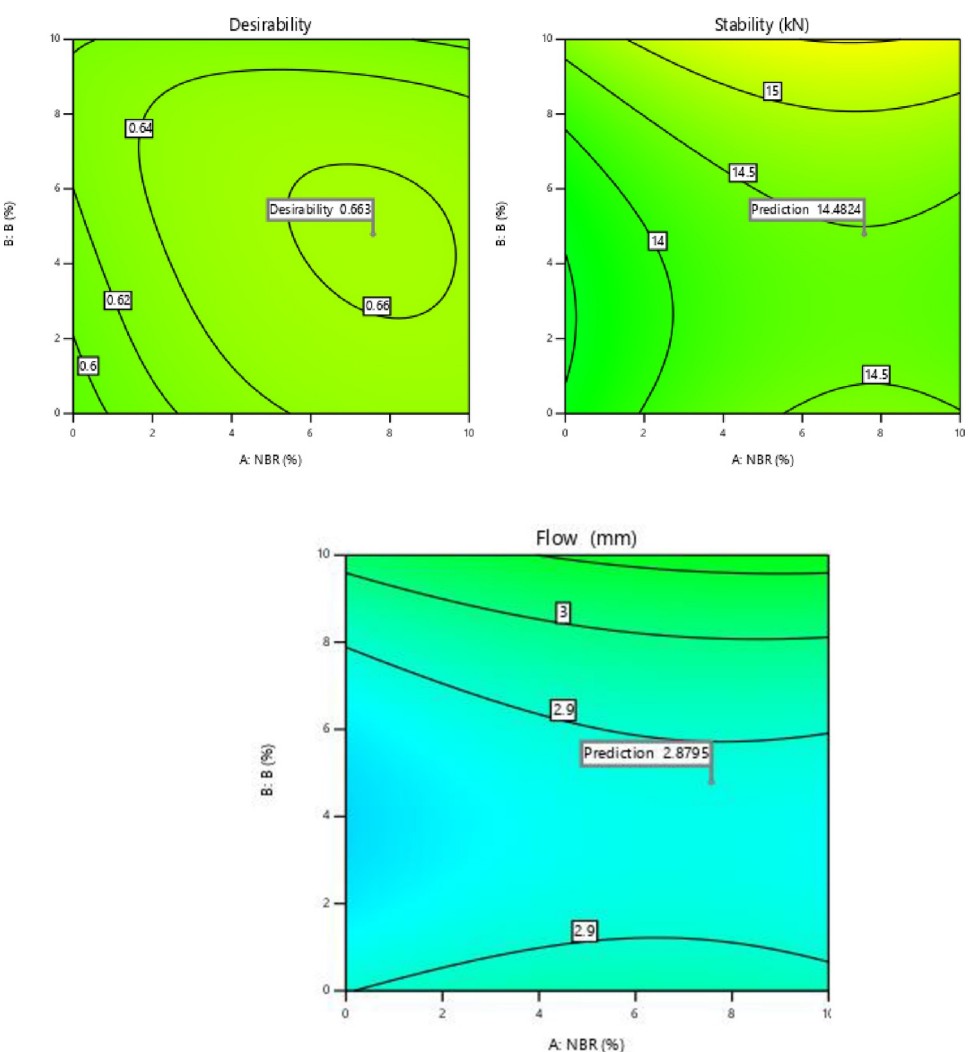

**Fig 8. Composite desirability of MS and FV.**

**Table 11.  Response during confirmatory experiments.**

| S.NO. | NBR (%) | B (%) | FR (%) | ER (%) | MS (kN) | | | FV (mm) | | |
|---|---|---|---|---|---|---|---|---|---|---|
| | | | | | Exp | Pred | % Error | Exp | Pred | % Error |
| 1 | 7.573 | 4.793 | 2.463 | 2.581 | 14.84 | 14.48 | 2 | 2.93 | 2.88 | 2 |
| 2 | 7.573 | 4.793 | 2.463 | 2.581 | 14.7 | 14.48 | 2 | 2.79 | 2.88 | 3 |
| 3 | 7.573 | 4.793 | 2.463 | 2.581 | 14.92 | 14.48 | 3 | 2.83 | 2.88 | 2 |
| 4 | 7.573 | 4.793 | 2.463 | 2.581 | 14.89 | 14.48 | 3 | 2.81 | 2.88 | 2 |
| 5 | 7.573 | 4.793 | 2.463 | 2.581 | 14.7 | 14.48 | 2 | 2.84 | 2.88 | 1 |

response parameters. The results of confirmation run for output responses are shown in Table 11.

After the confirmation test, it was found that the average values for MS and FV are 14.81 kN and 2.84 mm respectively, which also shows that the average percentage error is less than 5% which is a desirable characteristic.

## 7. Conclusions

The modified bituminous mix was successfully prepared by using modifiers NBR, Bakelite, FR and ER. The following points are summarized from the conducted research work.

i.  The regression model showed good correlation with the experimental results. The NBR, B, ER and FR are the major significant parameters for MS and FV.

ii.  Addition of Bakelite and ER increases the MS but decrease the FV. But by adding NBR and FR till intermediate value the MS increases and FV decreases, afterwards MS start declining and FV start rising.

iii.  It is clear from SEM and EDS analysis that sample S1having elements (5%-NBR, 10%-B, 10%-FR, 2.5%-ER) has better fine surface structure as compared to sample S34 (10%-NBR, 0%-B, 10%-FR, 2.5%-ER).

iv.  Multi Objective optimization using desirability approach suggests that the optimal conditions are obtained at NBR 7.6%, B 4.8%, FR 2.5% and ER 2.6% for MS and FV.

v.  Confirmation test conducted as per solution confirmed that the results for all output responses are within 5% error with optimal conditions.

## Supporting information

**S1 Appendix.**
(DOCX)

## Author Contributions

**Conceptualization:** Avani Chopra, Sandeep Singh, Abhishek Kanoungo, Gurpreet Singh, Naveen Kumar Gupta, Shubham Sharma.

**Formal analysis:** Avani Chopra, Sandeep Singh, Abhishek Kanoungo, Gurpreet Singh, Naveen Kumar Gupta, Shubham Sharma.

**Funding acquisition:** Shubham Sharma, Sayed M. Eldin.

**Investigation:** Avani Chopra, Sandeep Singh, Abhishek Kanoungo, Gurpreet Singh, Naveen Kumar Gupta, Shubham Sharma.

**Methodology:** Avani Chopra, Sandeep Singh, Abhishek Kanoungo, Gurpreet Singh, Naveen Kumar Gupta, Shubham Sharma.

**Project administration:** Shubham Sharma, Sayed M. Eldin.

**Supervision:** Shubham Sharma, Sanjeev Kumar Joshi, Sayed M. Eldin.

**Writing – original draft:** Avani Chopra, Sandeep Singh, Abhishek Kanoungo, Gurpreet Singh, Naveen Kumar Gupta, Shubham Sharma.

**Writing – review & editing:** Shubham Sharma, Sanjeev Kumar Joshi, Sayed M. Eldin.

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
