## [Decision Letter · Decision Letter 0]

28 Nov 2022

PONE-D-22-31489Multi-Objective Optimization of Nitrile Rubber and Thermosets Modified Bituminous Mix using Desirability Approach: Fabrication, and Evaluation of Marshall and Morphological characteristicsPLOS ONE

Dear Dr. Eldin,

Thank you for submitting your manuscript to PLOS ONE. After careful consideration, we feel that it has merit but does not fully meet PLOS ONE’s publication criteria as it currently stands. Therefore, we invite you to submit a revised version of the manuscript that addresses the points raised during the review process.

Please go through the recommendations of reviewers carefully and correct the manuscript accordingly.

We look forward to receiving your revised manuscript.

Kind regards,

Yasir Nawab, PhD

Academic Editor

PLOS ONE

Journal Requirements:

Reviewers' comments:

Reviewer's Responses to Questions

**Comments to the Author**

1. Is the manuscript technically sound, and do the data support the conclusions?

Reviewer #1: Yes

Reviewer #2: Yes

2. Has the statistical analysis been performed appropriately and rigorously? 

Reviewer #1: Yes

Reviewer #2: Yes

3. Have the authors made all data underlying the findings in their manuscript fully available?

Reviewer #1: Yes

Reviewer #2: Yes

4. Is the manuscript presented in an intelligible fashion and written in standard English?

Reviewer #1: Yes

Reviewer #2: Yes

5. Review Comments to the Author

Reviewer #1: There are many unnecessary references in the text that must be removed.

The manuscript has a poor style of writing.

The English grammar is also poor with many grammatical errors in the manuscript. The author(s) should correct the grammar.

Why are the abbreviations not stated fully at the time of first appearing in the manuscript?

The necessity of the research should be clearly stated in the abstract.

The method of doing the work should be clearly stated in the abstract.

The summary of the results should be stated at the end of the abstract.

The previous studies that have been done in recent years related to the topic of this research should be presented in the previous studies section.

The summary of previous studies and the necessity of the research should be stated.

The algorithm of the research method should be presented at the beginning of the laboratory program.

The results of this research should be investigated more deeply.

The results of this study should be compared with previous studies.

In the Conclusions section, it should be avoided to bring the results that were not specifically extracted from this research.

The description of the experiments should be stated more accurately and in detail.

The quality of the figures is very low.

The way to draw the figures should be more accurate.

Reviewer #2: Please refer to the comments as marked in the manuscript. In general, this manuscript requires major revisions before it can be considered for publication in PLOS ONE journal. Please do corrections carefully.

6. PLOS authors have the option to publish the peer review history of their article (what does this mean?). If published, this will include your full peer review and any attached files.

Reviewer #1: **Yes: **Gholam Hossein Hamedi

Reviewer #2: No

---

## [Author Response · Author response to Decision Letter 0]

11 Jan 2023

17.12.2022

Dear Prof. Dr. Editor-in-chief,

Thank you for considering the manuscript entitled, “Multi-Objective Optimization of Nitrile Rubber and Thermosets Modified Bituminous Mix using Desirability Approach: Fabrication, and Evaluation of Marshall and Morphological characteristics”, for publication in PLOS ONE (PLOSONE). I am grateful to you and the reviewers for the valuable suggestions provided. I like to resubmit our revised version of the manuscript by adding responses to all your queries. It has been analysing below for finding the answers and actions taken to address these comments. All the suggestions are incorporated and highlighted with the YELLOW COLOUR in the given manuscript. The locations of these changes have been mentioned as it was possible, in the action points that respond to each reviewer’s comments. Here are the responses to the reviewer’s comments:

AUTHOR RESPONSE TO REVIEWER AND EDITOR COMMENTS

Manuscript ID: PONE-D-22-31489

Paper title: Multi-Objective Optimization of Nitrile Rubber and Thermosets Modified Bituminous Mix using Desirability Approach: Fabrication, and Evaluation of Marshall and Morphological characteristics

RESPONSE TO REVIEWER’S COMMENTS

The authors are grateful to the reviewers for their suggestions that they all have contributed for improving the manuscript. Once again, the authors are extremely thankful for the observations and the comments of the reviewers. All the comments are appropriately addressed and now the quality of the article has been appreciably enhancing before the consideration for publication. The rebuttal file is enclosed indicating the revisions incorporated in the article as suggested. The revisions are carried out in YELLOW COLOUR in the text of the manuscript for better perusal to the reviewers, as well as for the editor. We have made the modifications as per their suggestions in the revised manuscript and changes are also marked up using YELLOW COLOUR.

All in all, the authors should thank the reviewers for their meticulous observations in reviewing the article. All the issues raised by the authors are appropriately addressed as stated following,

Reply to Reviewer’s comments

# Reviewer 1:

Reply-Thanks for the suggestion. The manuscript has been revised as per PLOS ONE style template.

2. There are many unnecessary references in the text that must be removed. The manuscript has a poor style of writing.

Reply-Thanks for the suggestion. The unnecessary references have been removed from the text in the revised manuscript draft.

3. The English grammar is also poor with many grammatical errors in the manuscript. The author(s) should correct the grammar.

Reply-Thanks for the suggestion. The manuscript has been revised and the English grammatical mistakes have been corrected.

4. Why are the abbreviations not stated fully at the time of first appearing in the manuscript?

Reply-Thanks for the suggestion. The abbreviations have been stated fully at the time of first appearing in the revised manuscript draft.

5. The necessity of the research should be clearly stated in the abstract. The method of doing the work should be clearly stated in the abstract. The summary of the results should be stated at the end of the abstract.

Reply-Thanks for the suggestion. The abstract has been revised as per the suggestions.

6. The previous studies that have been done in recent years related to the topic of this research should be presented in the previous studies section.

Reply-Thanks for the suggestion. The previous studies related to the topic of this research have been added in the previous studies section in the revised manuscript draft.

7. The summary of previous studies and the necessity of the research should be stated.

Reply-Thanks for the suggestion. The previous studies section has been updated with the summary and necessity of the research.

8. The algorithm of the research method should be presented at the beginning of the laboratory program.

Reply-Thanks for the suggestion. The flowchart has been added before the laboratory program in the revised manuscript draft.

9. The results of this research should be investigated more deeply.

Reply- Thanks for the suggestions, the results have been discussed deeply in the revised manuscript draft.

10. The results of this study should be compared with previous studies.

Reply-Thanks for the suggestion. The results of this study have been compared with the previous studies.

11. In the Conclusions section, it should be avoided to bring the results that were not specifically extracted from this research.

Reply-Thanks for the suggestion. The conclusion part has been revised as per suggestions in the revised manuscript draft.

12. The description of the experiments should be stated more accurately and in detail.

Reply- Thanks for the advice, the description of experiments has been stated clearly in the revised manuscript draft.

13. The quality of the figures is very low. The way to draw the figures should be more accurate.

Reply- Thanks for the advice. The figure quality has been enhanced in the revised manuscript draft.

14. Results and Discussion sections are missing.

Reply- Thanks for the valuable suggestion, the results and discussion section has been added and results have been discussed in detail using individual plots, actual vs predicted plots, and interaction plots in the revised manuscript draft.

# Reviewer 2:

1. Title is too long. Maybe the words "Fabrication and Evaluation.....until the end" can be taken out.

Reply-Thanks for the suggestion. The title has been revised as per suggestions.

2. Normally we used the term like this " a 60/70 penetration grade bitumen".

Reply-Thanks for the suggestion. The term " a 60/70 penetration grade bitumen" has been rewritten in the revised manuscript draft.

3. Figure 2 is not necessary and can be taken out from the manuscript.

Reply-Thanks for the suggestion. Figure 2 has been removed from the manuscript.

4. Where are others OBC? Did you just use one OBC for all the different mixtures?

Reply-Thanks for the suggestion. The Optimum Bitumen Content is obtained for the Conventional Mix (without a modifier) and the same value of OBC has been considered for the DOE.

5. This figure is not necessary and can be taken out from manuscript.

Reply-Thanks for the suggestion. The figure has been removed from the manuscript.

6. Can we put Table 9 in the Appendix? As the table is too long. 

Reply-Thanks for the suggestion. Table 9 has been put in the Appendix.

7. The authors only reported what had been done in this study. They should compare and discuss their findings with previous studies done by other researchers. Please revise the whole Results and Discussion thoroughly.

Reply-Thanks for the suggestion. The Results and Discussions section has been revised by adding more results in the revised manuscript draft.

8. Table 10 and 11 is okay to be put in a thesis, but not necessary for a journal paper. It is recommended that the authors have an Appendix section for this paper.

Reply-Thanks for the suggestion. The table 10 and 11 have been put in the Appendix as per recommendation.

9. Combine Figures 4 and 5 as one figure only.

Reply-Thanks for the suggestion. Figures 4 & 5 have been combined in the revised manuscript draft.

10. Combine Figures 6 and 7 as one figure only.

Reply-Thanks for the suggestion. Figures 6 & 7 have been combined in the revised manuscript draft.

11. Are the two equations important to be shown in the paper?

Reply-Thanks for the suggestion. The equations are indicators of the Multi-objective Optimization using equal weights so have been added to the paper.

12. Please rewrite the desirability function in the paragraph.

Reply-Thanks for the suggestion. The paragraph has been rewritten in the revised manuscript draft.

Thus, a scientific explanation of the obtained results has been refined and ameliorated up to a fervent extent. Results are enumerated, test methods are utterly described, and interpretations have been correlated with results and previous literature findings. The overall summary should indicate the progress of the research and its limitations.

Note: All the necessary changes/added sentence has been shown in YELLOW COLOUR.

Thank you very much in advance for taking your time in reviewing this manuscript.

Sincerely, we hope you will find our revision satisfactory.

Thanks, in anticipation.

Regards,

Dr. Shubham Sharma

(Corresponding author)

---

## [Decision Letter · Decision Letter 1]

24 Jan 2023

Multi‑Objective Optimization of Nitrile Rubber and Thermosets Modified Bituminous Mix using Desirability Approach

PONE-D-22-31489R1

Dear Dr. Eldin,

We’re pleased to inform you that your manuscript has been judged scientifically suitable for publication and will be formally accepted for publication once it meets all outstanding technical requirements.

Kind regards,

Yasir Nawab, PhD

Academic Editor

PLOS ONE

Additional Editor Comments (optional):

Reviewers' comments:

Reviewer's Responses to Questions

**Comments to the Author**

1. If the authors have adequately addressed your comments raised in a previous round of review and you feel that this manuscript is now acceptable for publication, you may indicate that here to bypass the “Comments to the Author” section, enter your conflict of interest statement in the “Confidential to Editor” section, and submit your "Accept" recommendation.

Reviewer #1: All comments have been addressed

Reviewer #2: All comments have been addressed

2. Is the manuscript technically sound, and do the data support the conclusions?

Reviewer #1: Yes

Reviewer #2: Partly

3. Has the statistical analysis been performed appropriately and rigorously? 

Reviewer #1: Yes

Reviewer #2: Yes

4. Have the authors made all data underlying the findings in their manuscript fully available?

Reviewer #1: Yes

Reviewer #2: Yes

5. Is the manuscript presented in an intelligible fashion and written in standard English?

Reviewer #1: Yes

Reviewer #2: Yes

6. Review Comments to the Author

Reviewer #1: (No Response)

Reviewer #2: The authors have addresses all comments given by the reviewers. Therefore, the current form of this manuscript is okay and can be accepted for publication in PLOS ONE.

7. PLOS authors have the option to publish the peer review history of their article (what does this mean?). If published, this will include your full peer review and any attached files.

Reviewer #1: **Yes: **Gholam Hossein Hamedi

Reviewer #2: No

---

## [Editor Report · Acceptance letter]

27 Jan 2023

PONE-D-22-31489R1 

Multi‑Objective Optimization of Nitrile Rubber and Thermosets Modified Bituminous Mix using Desirability Approach 

Dear Dr. Eldin:

I'm pleased to inform you that your manuscript has been deemed suitable for publication in PLOS ONE. Congratulations! Your manuscript is now with our production department. 

Kind regards, 

on behalf of

Dr. Yasir Nawab 

Academic Editor

PLOS ONE